# A Genomic Approach to Delineating the Occurrence of Scoliosis in Arthrogryposis Multiplex Congenita

**DOI:** 10.3390/genes12071052

**Published:** 2021-07-08

**Authors:** Xenia Latypova, Stefan Giovanni Creadore, Noémi Dahan-Oliel, Anxhela Gjyshi Gustafson, Steven Wei-Hung Hwang, Tanya Bedard, Kamran Shazand, Harold J. P. van Bosse, Philip F. Giampietro, Klaus Dieterich

**Affiliations:** 1Grenoble Institut Neurosciences, Université Grenoble Alpes, Inserm, U1216, CHU Grenoble Alpes, 38000 Grenoble, France; XMartin@chu-grenoble.fr; 2Shriners Hospitals for Children Headquarters, Tampa, FL 33607, USA; SCreadore@shrinenet.org (S.G.C.); agustafson@shrinenet.org (A.G.G.); kshazand@shrinenet.org (K.S.); 3Shriners Hospitals for Children, Montreal, QC H4A 0A9, Canada; ndahan@shrinenet.org; 4School of Physical & Occupational Therapy, Faculty of Medicine and Health Sciences, McGill University, Montreal, QC H3G 2M1, Canada; 5Shriners Hospitals for Children, Philadelphia, PA 19140, USA; sthwang@shrinenet.org (S.W.-H.H.); hvanbosse@shrinenet.org (H.J.P.v.B.); 6Alberta Congenital Anomalies Surveillance System, Alberta Health Services, Edmonton, AB T5J 3E4, Canada; tanya.bedard@albertahealthservices.ca; 7Department of Pediatrics, University of Illinois-Chicago, Chicago, IL 60607, USA; 8Institut of Advanced Biosciences, Université Grenoble Alpes, Inserm, U1209, CHU Grenoble Alpes, 38000 Grenoble, France

**Keywords:** Amyoplasia, scoliosis, DECIPHER (DatabasE of genomiC variation and Phenotype in Humans using Ensemble Resources), CNV (copy number variant), DA (distal arthrogryposis), IPA (ingenuity pathway analysis), HPO (human phenotype ontology), akinesia, MYOD, IGF2

## Abstract

Arthrogryposis multiplex congenita (AMC) describes a group of conditions characterized by the presence of non-progressive congenital contractures in multiple body areas. Scoliosis, defined as a coronal plane spine curvature of ≥10 degrees as measured radiographically, has been reported to occur in approximately 20% of children with AMC. To identify genes that are associated with both scoliosis as a clinical outcome and AMC, we first queried the DECIPHER database for copy number variations (CNVs). Upon query, we identified only two patients with both AMC and scoliosis (AMC-SC). The first patient contained CNVs in three genes (*FBN2*, *MGF10*, and *PITX1)*, while the second case had a CNV in *ZC4H2*. Looking into small variants, using a combination of Human Phenotype Ontogeny and literature searching, 908 genes linked with scoliosis and 444 genes linked with AMC were identified. From these lists, 227 genes were associated with AMC-SC. Ingenuity Pathway Analysis (IPA) was performed on the final gene list to gain insight into the functional interactions of genes and various categories. To summarize, this group of genes encompasses a diverse group of cellular functions including transcription regulation, transmembrane receptor, growth factor, and ion channels. These results provide a focal point for further research using genomics and animal models to facilitate the identification of prognostic factors and therapeutic targets for AMC.

## 1. Introduction

Arthrogryposis multiplex congenita (AMC or interchangeably arthrogryposis) describes a group of conditions characterized by the presence of non-progressive congenital contractures in multiple body areas [1]. The congenital contractures are a result of decreased fetal movement (fetal akinesia), leading to joint fibrosis and dysplasia/lack of elasticity of the soft tissues surrounding the joint. The longer the duration and the earlier the onset of fetal akinesia, the more severe the contractures. The direct causes of fetal akinesia are varied, including abnormalities of the gross or microscopic neurologic system (from brain architecture to anterior horn cell formation), abnormalities of muscle function, restrictive connective tissue conditions, as well as intrauterine crowding and maternal disease. There are more than 400 underlying conditions identified that can lead to fetal akinesia and subsequently to a baby born with AMC, and most of these conditions have known genetic causes [2,3]. Due to the heterogeneity of the condition, several classification systems for AMC exist [1,4,5]. The classification system by Bamshad et al. separates the AMC types by their cardinal features, creating three categories of a roughly equal number of cases. The first category consists of only one diagnosis, Amyoplasia, which represents approximately one-third of cases of arthrogryposis. This “classic arthrogryposis” is a distinct clinical entity presenting with hypoplasia or atrophy of specific muscle groups, and multiple joint contractures. Features at birth are very recognizable and include internal rotation and adduction of shoulders, extended elbows, flexed wrists, and equinovarus foot deformities [1]. Dimples over affected joints are evident. Other recognizable features include a lack of flexion creases on hands and nevus flammeus over the forehead. No underlying genetic abnormality or family history has been associated in cases with Amyoplasia. Therefore, Amyoplasia is postulated to have nongenetic causes, with an intra-uterine vascular interruption as the leading hypothesis [6]. The second category consists of the distal arthrogryposes (DAs), defined by the presence of congenital contractures of primarily the distal joints, primarily wrist and hand contractures and foot deformities (clubfoot or congenital vertical talus), but also to a lesser extent elbows, knees, shoulders, and hips. Underlying genetic causes have been described in most DAs. The current classification system for DA includes 11 subtypes, but as many as 19 different DAs have been suggested. The third category, Bamshad’s syndromic category, is used to denote cases of arthrogryposis which may be associated with bone or central nervous system involvement and other birth defects or malformations. This category will probably undergo substantial reorganization in the coming years, as the similarities between different conditions become better understood and the underlying molecular causes unveiled.

The prevalence of scoliosis in children with AMC has been variably reported between 20% and 66%, although more recent studies place the prevalence in the range of 20 to 25% [7,8,9,10]. Scoliosis is defined as a coronal plane spine curvature of 10 degrees or greater as measured radiographically and can be separated into idiopathic, congenital, and syndromic or neuromuscular scoliosis. Idiopathic scoliosis represents a curvature of the spine for which no definitive underlying cause is yet known, although a number of candidate genes have been identified [11,12]. Congenital scoliosis is caused by vertebral malformations such as failure of formation (hemivertebrae) and/or failure of segmentation (congenital fusion of two or more vertebral levels). Very few arthrogrypotic conditions will have associated congenital vertebral malformations, therefore most cases of AMC-associated scoliosis are syndromic or neuromuscular. While most published series of children with AMC and scoliosis (AMC-SC) are relatively small, ranging from 14 [7] to 117 patients [9], the relatively high rate of spinal involvement is notable when assessing and treating children with AMC. Some types of AMC have a high association with scoliosis, while others rarely develop spinal deformities. Since most underlying conditions of AMC have known genetic causes, understanding these conditions could shed a light on pathways leading to scoliosis related to arthrogryposis [2,3]. 

Our primary goal for this paper was to characterize the genetics of the AMC types that have a strong association with scoliosis. We undertook a systematic review of all known genes associated with AMC, focusing on those with an association with scoliosis. We also analyzed copy number variants (CNVs) which represent structural variations in chromosome regions associated with duplication and deletion of genomic material, for their possible role in arthrogryposis and scoliosis. By delineating genes associated with both conditions, common pathways and potential mechanisms were identified to improve our understanding of the natural history of some forms of arthrogryposis, provide prognostic information for health care providers and families caring for children with arthrogryposis, and possibly lead to targeted therapies for affected patients. 

## 2. Materials and Methods

To delineate the genes associated with both AMC and scoliosis (AMC-SC), their common pathways, and potential mechanisms, we first identified the genes associated with AMC as well as the genes associated with scoliosis. We then identified the common genes to both sets of conditions and conducted Gene Interaction Pathway Analysis, followed by an identification of the CNVs for the identified genes. Each of these steps are detailed below.

### 2.1. Identification of Genes Associated with AMC

Two previous gene ontology articles published in 2016 and 2019 established a group of 402 genes associated with AMC, which were used as the initial source to identify the genes associated with AMC [2,3]. In addition, we consulted the literature in PubMed from 2019 until 31 December 2020, to identify additional genes since 2019 that are associated with AMC. We identified 30 additional genes (see Appendix A) resulting in a total of 444 AMC-associated genes (Figure 1).

Simultaneously, we extracted 112 genes associated with AMC using the Human Phenotype Ontology (HPO) project (identifier HP:0002804; accessed on Tuesday April 13th 2021, version hpo-web@1.7.9-hpo-obo@2021-02-08). Of these 112 genes, 94 were already identified by the literature search, and the remaining 18 genes were manually curated through a literature review for association with AMC, only 12 of which were found to be associated with AMC. These 12 genes were: *C12orf65*, *DSE, NEK9, PHGDH, PPP3CA*, *PSAT1*, *TBCD*, *VAMP1*, *CACN1E*, *CEP55, RFT1*and *SHPK.*

### 2.2. Identification of Genes Associated with Scoliosis 

The same method used for AMC was applied to identify the genes associated with scoliosis using the Human Phenotype Ontology (HPO) project (identifier HP:0002650; accessed on Tuesday April 13th 2021, version hpo-web@1.7.9-hpo-obo@2021-02-08). A total of 895 genes associated with scoliosis extracted using HPO were reviewed. An additional 16 genes reported in the literature based on Perez-Machado and colleagues’ 2020 paper since then were also reviewed, of which three were duplicates among the 895 genes already identified, resulting in a total of 908 genes (see Figure 1) [12].

### 2.3. Identification of Genes Associated with Both AMC and Scoliosis and Gene Interaction Pathway Analysis

The list of genes identified for AMC and for scoliosis were compared to identify the genes that are associated with both. Ingenuity Pathway Analysis (IPA), which represents a functional analysis of a set of identified genes, was then conducted using the IPA Ingenuity Systems QIAGEN, content version 60467501 software. A core analysis type and subsequent variant effect analysis were used to generate the outputs in each case. 

### 2.4. Identification of Copy Number Variants (CNV) Associated with AMC and Scoliosis

In order to identify CNVs associated with both AMC and scoliosis, we queried the DECIPHER (DatabasE of genomiC variation and Phenotype in Humans using Ensemble Resources) (https://decipher.sanger.ac.uk/) database to identify reported cases with scoliosis or vertebral malformation(s) with AMC (accessed on 9 February 2021). To do so, the 444 genes associated with AMC were queried through the implementation of an in-house Selenium-based automation software package written in the Python 3.8 programming language. The data points extracted into DECIPHER included gene name, number of associated genes, DECIPHER patient number, phenotype(s)/conditions, chromosome location, start position, end position, mode of inheritance, and genotype. The resulting DECIPHER patient IDs with their associated data were then sub-sampled into identifiable cohort groups representing the phenotype(s) of interest including Arthrogryposis-like hand anomaly, Arthrogryposis Multiplex Congenita, Distal Arthrogryposis, and Scoliosis. We then filtered for DECIPHER Patient IDs containing both arthrogryposis and scoliosis. The arthrogryposis and scoliosis DECIPHER Patient IDs containing single nucleotide variants (SNVs) were removed by subtracting the start position from the end position to identify the allelic depth and kept only copy number variation (CNV). Next, we removed any duplicates within our dataset resulting in an accurate representation of the copy number variant genes associated with arthrogryposis and scoliosis for the DECIPHER Patient IDs extracted.

## 3. Results

Combining the initial literature search results with the HPO identified genes for AMC-SC independently yielded a total of 444 genes associated with AMC and 908 genes associated with scoliosis. When comparing these two sets of genes, 227 genes were found in common (Figure 2). 

This set of 227 genes was then analyzed using IPA. Overall, this group of 227 genes encompasses a diverse group of cellular functions including transcription regulation, transmembrane receptors, growth factor-related genes, and ion channels. (Table 1). 

Figure 3A shows examples of the 227 genes common to scoliosis and AMC analyzed in IPA. On the canonical pathway panel (A, left) describing the actin cytoskeletal pathway, major disruption points representing inactivating variants in some key genes such as F-actin, Myosin, and Filamin A (FLNA) that crosslinks actin filaments to membrane glycoproteins can be seen. 

As of the right panel (B), the complex intricate interaction network clearly shows the close functional relationship and involvement of key genes such as AKT Serine/Threonine Kinase 1 (Akt), cholinergic receptor family (CHRN), and Neuregulin gene family (NRG) involved in neuromuscular junctions. 

The DECIPHER database search (outlined in Figure 4) identified only two patients harboring CNVs associated with scoliosis and arthrogryposis, for which details are summarized in Table 2.

The first patient (#260667) had a chromosome 5 deletion encompassing 133 genes. Three relevant genes contained within the CNV which potentially impacted the phenotype i.e., *FBN2*, *MEGF10,* and *PITX1* [46]. The CNV is a 10.82 Mb heterozygous deletion containing 133 genes resulting in a contiguous gene deletion syndrome. This deletion has been documented with a haploinsufficiency score of 50.51, i.e., a high likelihood of causing a loss of function [47].

Mono-allelic *FBN2* mutations are associated with Beals congenital contractural arachnodactyly [48]. Bi-allelic mutations in *MGF10* are associated with myopathy, areflexia, respiratory distress, and dysphagia. Mono-allelic mutations in *PITX1* are associated with congenital clubfoot, with or without deficiency of long bones and/or mirror-image polydactyly in addition to Liebenberg syndrome, defined by the presence of carpal synostosis, dysplastic elbow joints, and brachydactyly [49].

Regarding patient #2, the de novo heterozygous CNV is a fragment of 233.15 Kb located on the X chromosome, containing only the 2C4H2 gene and reported as “likely pathogenic” according to the ClinVar classification.

The second patient (#262492) had a heterozygous or hemizygous (on the X chromosome) deletion encompassing the *ZC4H2* part of the CNV. *ZC4H2* is associated with Wieacker–Wolff syndrome, characterized by the presence of foot contractures, muscle atrophy, and oculomotor apraxia [50].

## 4. Discussion

Despite the significant prevalence of scoliosis in the AMC patient population, there is little information regarding genetic contributions to scoliosis development in AMC in the literature. In one study of 46 patients with AMC, 32 patients (65.6%) developed scoliosis between the ages of 5–16 years [10]. Five of 32 patients (15.7%) presented with scoliosis at birth, reflecting the position of the immobile fetus in the uterus, and therefore referred to as “prenatal scoliosis”. Several patterns of scoliosis have been noted to occur in AMC and include a “paralytic curve” which is more common in the hypotonic types of AMC and tends to progress; it is typically observed before the age of 2 years but can arise at any age. These curves tend to be thoracolumbar in local, often with pelvic obliquity and severe hip contractures. The second curve pattern is the less prevalent “idiopathic-like”, with more balanced double curves, localized to the thoracic and thoracolumbar regions, and often manifesting in later childhood or adolescence. 

The aims of this review were to identify genes that are associated with both AMC and scoliosis, and describe the functional pathways and CNVs associated with both conditions. While additional analysis of comparison between pathways associated with arthrogryposis without scoliosis, scoliosis without arthrogryposis and pathways that are associated with both arthrogryposis and scoliosis may be potentially complementary this was not the ultimate focus of our investigation. To our knowledge, this is the first study that has utilized a genomic approach to identify genes that are associated with both AMC and scoliosis. We identified a list of 227 genes that were associated with AMC-SC. The collection of genes encompasses a diverse group of cellular functions, which, once impaired, contribute to AMC-SC: cytoskeletal elements, neurotransmitter enzyme function, molecular chaperone function, ion channel regulation, extracellular matrix, DNA repair, growth factor, transmembrane receptor, transcription factor/regulator, messenger RNA regulation and cellular transport (see Table 1). 

IPA analysis of the 227 AMC-SC genes suggest some common causal mechanisms and pathways such as critical “housekeeping” functions (cellular ion balance, DNA excision/repair, mRNA trafficking and post-translational modification), embryologic development, and structural families of genes expressed in bones and/or muscles (Figure 3A,B). The affected genes can heavily impair the naturally occurring or canonical pathways at crucial points, degrading the normal progression of embryologic development and/or after birth differentiation. This process depends on the chronological expression of involved genes and their transcriptional factors (TFs). Several guiding principles were demonstrated by the intricate relationships of the studied genes, as visualized by the pathway analysis: (1) It is likely that the majority of these genes are related to the pathological processes involved with the development of arthrogryposis and scoliosis, (2) IPA analysis facilitates a birds-eye view of potentially impaired key processes, (3) In the central nervous system, dysfunction can be related to mutations of genes surrounding the AKT1/2 kinases or of the growth factors regulating their activities, neurotransmitter receptors, or intracellular ion balance impairing the transmission of electrical impulses, (4) Mutations in structural genes such as actin, myosin, titin, and dynein in bone and muscle related pathways may cause impaired cytoskeletal function and/or decreased contractile ability, (5) Mutations in regulatory genes such as *TBX5*, *TRIP4* or *NFkB* act at the level of transcription to regulate activity of these genes, (6) There are inflammatory mediated effects on cellular differentiation in these organs. As an example of how the IPA analysis can reflect actual findings, the analysis attests an interaction between MYH3 and actin (Figure 3A). Mutations in *MYH3* are associated with Freeman-Sheldon syndrome (FSS), a form of DA characterized by a small mouth and joint contractures. Drosophila modeling of FSS provided molecular evidence for MYH3 and actin interaction as *MYH3* mutations are associated with myofibrillary disarray and result in decreased catalytic efficiency of actin-activated ATP hydrolysis [51]. IPA analysis did identify other potential disorders and conditions that may be attributed to alterations in genes which are members of pathways in which the 227 genes identified with AMC-SC. These include skeletal, muscular, limb defects and cognitive disability. Further validation would require a more in depth analysis, which is not the focus of this paper.

Figure 3B highlights among other interactions, interplay between *MYOD1* and *IGF2*. Literature review supports this interaction. Recently, two siblings presenting with a lethal form of fetal akinesia deformation sequence (FADS) including deficient pectoralis and proximal limb musculature, distal joint contractures and neonatal respiratory have been described. Watson et al. [52] found a homozygous probably pathogenic loss of function variant, c.188C>A/ p.Ser63*, in *MYOD1*. *MYOD1* encodes MyoD. MyoD is a key player in cell proliferation and differentiation of myoblasts and its expression fine tunes the balance between myoblast proliferation and differentiation [53]. MyoD directly activates the expression of a long non coding mRNA, called LncMYOD, encoded next to the *MYOD1* gene [54]. LncMyoD then interacts directly with IMP2 (insulin-like growth factor 2 mRNA-binging protein 2). LncMyoD downregulates IMP2-mediated mRNA translation of genes involved in cell proliferation, such as *N-RAS* and c-myc and *IGF2*. Interestingly *IGF2* is part of the imprinting control region 1 (ICR1) at chromosome 11p15.5. *IGF2*, as well as the *H19* gene, when hypomethylated at the ICR1 locus, are associated with Silver-Russell syndrome [55]. Patients with Silver-Russell syndrome have major clinical features consistent with pre- and postnatal growth restriction, frontal bossing with relative macrocephaly, feeding difficulties and low body mass index. In some individuals musculoskeletal features have also been mentioned with muscle hypoplasia and congenital joint contractures [56,57].

Querying the DECIPHER database yielded only two patients who had CNV associated with AMC-SC (Figure 4 and Table 2). Patient 1 had a much larger region of CNV, containing three relevant genes: *FBN2*, *MEGF10* and *PITX1* [46]. *FBN2* (Fibrillin 2) codes for cytoskeletal element, and mutations are associated with Beals congenital contractural arachnodactyly [48]. *FBN2* intragenic deletions or splice site mutations have been published on some occasions associated with contractural congenital arachnodactyly [58]. Rare nonsense mutations are present in the ClinVar database and reported as pathogenic. These latter are not known to be associated with an AMC phenotype. Therefore we cannot either exclude or confirm a link between *FBN2* haploinsufficiency and AMC *MEGF10* (multiple epidermal growth factor-like domains protein 1) codes for a membrane receptor involved in the phagocytosis of apoptotic cells by macrophages and astrocytes, and biallelic mutations are associated with myopathy, areflexia, respiratory distress and dysphagia. *PITX1* (Paired Like Homeodomain 1) was the only gene of the CNV analysis that had not also been identified as a gene associated with AMC-SC, and in the literature, it has not yet been associated with AMC. It is associated with congenital clubfoot, occasionally with bony malformations of the foot, and Liebenberg syndrome, defined by the presence of carpal synostosis, dysplastic elbow joints and brachydactyly [49,59]. Other nonsense variants (2) have only been mentioned in ClinVar.

It is unclear if the monoallelic *MEGF10* was responsible for any part of the patient’s phenotype of either AMC or scoliosis. There is a possibility *FBN2*, a gene known to cause Beals syndrome (a form of distal arthrogryposis with scoliosis), was the only gene responsible for the AMC-SC phenotype; and there is a possibility of some type of additive effect between monoallelic *FBN2*, *MEGF10*, and *PITX1* resulting in an arthgrogrypotic phenotype. We suspect this patient has a contiguous gene syndrome. Additional literature reports describing patients with similar phenotypic features and deletions would provide support for this hypothesis.

One microdeletion involving only *PITX1* has been associated in one family with clubfoot over three generations (Alvarado et al., 2011 [59]). Other nonsense variants (2) have only been mentioned in ClinVar.

Patient 2 had a CNV for *ZC4H2* (Zinc Finger C4H2-Type Containing), a gene causing an X-linked arthrogryposis, usually only in females, known as Wieacker-Wolff syndrome or *ZC4H2*-Associated Rare Disease (ZARD). This condition is characterized by hypotonia, moderate to severe developmental delay, and early and progressive onset of scoliosis. Loss of function mutations have been described to be pathogenic on several occasions in *ZC4H2* [50,60,61,62]. 

It is of interest to notice the small number of mapped CNVs in patients as opposed to coding region variants in the literature. This can be at least partially explained by a historic bias of research toward the exome. With a current increase in whole genome sequencing clinical projects, we should see an increase in the non-coding variants for all disease-related literature in the next few years. Furthermore, geneticists do not necessarily need to submit CNVs to DECIPHER that can be easily linked to clinical symptoms. 

Future directions related to this research could be exploration of possible treatments, either to ameliorate the effects of AMC and possibly avoid the scoliosis, or to prevent the fetal akinesia altogether. IPA analysis has in some instances the ability to suggest possible drug targets for specific gene pathways. For example, many of the genes implicated in fetal akinesia also are associated with later life cancers. A number of medications have been used or are being developed to treat these cancers. Viewing these drugs as therapeutic for AMC must be done with extreme caution. Mechanistically, many of the genetic AMCs are due to lack of function or altered function of the gene product, whereas many cancers are often an uncontrolled overexpression of the proliferative genes (also shut off of differentiation genes) IPA analysis is limited with respect to drugs’ interactions and genes or gene products. While IPA may suggest a certain drug for its interaction with a particular gene or gene product, the nature of the interaction could be unclear. For instance, curarizing agents are listed with *CHRNG* which codes for a subunit of the acetylcholine receptor (AChR), and a mutation of which is associated with multiple pterygium syndrome (MPS). But mutations of *CHRNG* that cause MPS are loss of function mutations which result in a failure of export of the subunit to the cell surface or no protein expression [63]. The mutation already has a disconnecting effect on the neuromuscular junction, which curarizing agent would only exacerbate. Additionally, the *CHRNG* encoded protein is only expressed up to the 33rd week of pregnancy, and is replaced on the AChR with an “adult” subunit, which is presumably functional. In fact, it should be assumed that any chemical treatment for underlying causes of AMC-SC would need to be given early in pregnancy in order to prohibit development of deformities related to fetal akinesia.

Research using animal models such as zebrafish has shown some promise in the identification of pathologic mechanisms which may be amenable to targeted therapies. *MYBPC1* appears to be a novel gene responsible for DA1, though the mechanism of disease may differ from how some cardiac *MYBPC3* mutations cause hypertrophic cardiomyopathy [64]. *MYBPC1* is necessary in slow skeletal muscle development and can be used in established zebrafish models as a tractable model of human distal arthrogryposis [65]. Mutations in *MYH3*, which encodes embryonic heavy chain (MyHC) expressed initially during slow skeletal muscle development are also associated with multiple pterygium syndrome (MPS) and spondylocarpotarsal synostosis syndrome. The latter condition is characterized by joint contractures in addition to vertebral, carpal and tarsal fusions, and could present a mechanistic link between vertebral fusions and joint contractures, with hypercontraction of the surrounding muscle leading to excessive notochord tension [66,67]. Zebrafish homozygous for the smyhc (slow myosin heavy chain) are analogous to the most common distal arthrogryposis caused by MYH3 mutations. The zebrafish develop notochord kinks characterized by vertebral fusions, progression to scoliosis in addition to motor deficits accompanied the disorganized and shortened slow-twitch skeletal muscle myofibers. Slow twitch muscle fibers rely on aerobic metabolism and are recruited for smaller range of activities as compared to fast twitch fibers which rely on anaerobic metabolism and are utilized for larger bursts of activity. Treatment of the zebrafish embryos with the myosin ATPase inhibitor, para-aminoblebbistatin, which decreases actin-myosin affinity, normalized the vertebral fusions and notochord phenotype [68]. These findings hold tremendous promise for the treatment of AMC-SC.

*TNNI2* is also associated with distal arthrogryposis, types DA1 and DA2B, encoding a subunit of the troponin complex. Tnni2^K175del^ transgenic mice with a heterozygous gain of function mutation in *TNNI2*, encoding a subunit of the troponin complex have small body size and joint contractures. Hypoxia-inducible factor3a (Hif3a) was found to be increased with decreased Vegf levels in bone in these mice resulting in decreased angiogenesis, delays in endochondral ossification, decreased chondrocyte differentiation and osteoblast proliferation [69]. Interestingly, both *HIF3A* silencing using Hif3A/Hif-3α siRNA and HIF-prolyl hydroxylase inhibition effectively increased the cell viability during anoxia/reoxygenation injury in cardiomyocytes and led to changes in mRNA expression of HIF1-target genes, and reduced the loss of mitochondrial membrane potential (Δψ_m_) [70]. These results show promise towards applications for AMC bone related targeted treatments.

We noted a number of methodological barriers in our research. Although similar database and literature searches were implemented to identify scoliosis- and AMC-associated genes (see Figure 1), it is noteworthy that HPO scoliosis-associated terminology accounted for 98.5% of scoliosis-associated genes, with literature review adding only another 1.5%, whereas using HPO terminology for AMC identified only 21.7% of AMC-associated genes, lagging significantly compared to the literature search. There are multiple reasons to use the HPO terminology to identify genes associated with AMC-SC. HPO is freely available, provides standardized vocabulary for phenotypic manifestations of genetic disorders and can aid in specific diagnoses. HPO also provides linkages with different disease coding systems. The lack of AMC-associated terminology in HPO stems from the scarcity of codes associated with rare AMC-associated disorders, particularly those conditions categorized under Bamshad’s syndromic category [71]. Since its founding in 2008, HPO continues to expand its coverage of disease-associated phenotypes [72]. However, some disorders such as epilepsy have very deep phenotypic characterization, whereas other disorders such as respiratory diseases are less well represented in HPO. In searching the HPO database with AMC-associated terms, 18 genes were singled out as not having been identified in the literature search (see Figure 1); subsequently 6 were discarded as on further review as they did not have an associated AMC phenotype. 

We strived to be comprehensive in this study, with the identification of genes associated with AMC-SC using PubMed, HPO, and DECIPHER. Despite those efforts, genes may have been missed or their phenotypic spectrum not completely realized, particularly those in research or other databases that are not public/accessible and have not been published. For instance, *MYH3* is the only gene associated with Freeman-Sheldon syndrome (and the only gene associated with DA8–autosomal dominant MPS). We would suggest further investigation into the specific gene mutations of *MYH3* that leads to the occurrence of scoliosis. *MYH3* mutations can also lead to DA1, the “classic” distal arthrogryposis, as well as DA2B, Sheldon-Hall syndrome, both of which rarely have associated scoliosis. Understanding the differences in the specific mutations of *MYH3* between these three conditions may shed light on the origins of scoliosis.

Ascertainment bias or failure to include patients with AMC or scoliosis could lead to a misrepresentation of the number of the genes associated with AMC-SC. We used a systematic HPO and literature searching approach to identify genes associated with AMC, scoliosis so we could subsequently identify genes associated with both conditions.

Syndromes/genes associated with AMC-SC are relatively rare, which highlights the need to share findings and contribute to more easily accessible platforms such as HPO. An ongoing registry project for children with AMC, funded by the Shriners Hospitals for Children and implemented in seven regions in North America, will make future contributions to genotype/phenotype associations in AMC. This should lead to a better understanding of mechanisms that lead to AMC, and possibly to better care and outcomes. International collaborations to expand this registry have started and will necessitate the identification of common data elements and terms. There will also be opportunities for the findings from this registry to contribute to such platforms as HPO.

## 5. Conclusions

Using a combination of HPO analysis and literature review, we identified 908 genes associated with scoliosis and 444 genes associated with AMC resulting in 227 genes associated with AMC-SC. These genes act through a variety of cellular mechanisms including transcription regulation, transmembrane receptor, growth factor, and ion channels. Through query of the DECIPHER database, we identified two patients each with one CNV associated with AMC-SC. The first case had a CNV involving three genes (*FBN2*, *MEGF10,* and *PITX1)*, while the second case had a CNV involving *ZC4H2*. As we continue to learn more about genetic mechanisms responsible for AMC we anticipate the ability to better provide prognostic information and targeted therapies for affected patients.

## Figures and Tables

**Figure 1 genes-12-01052-f001:**
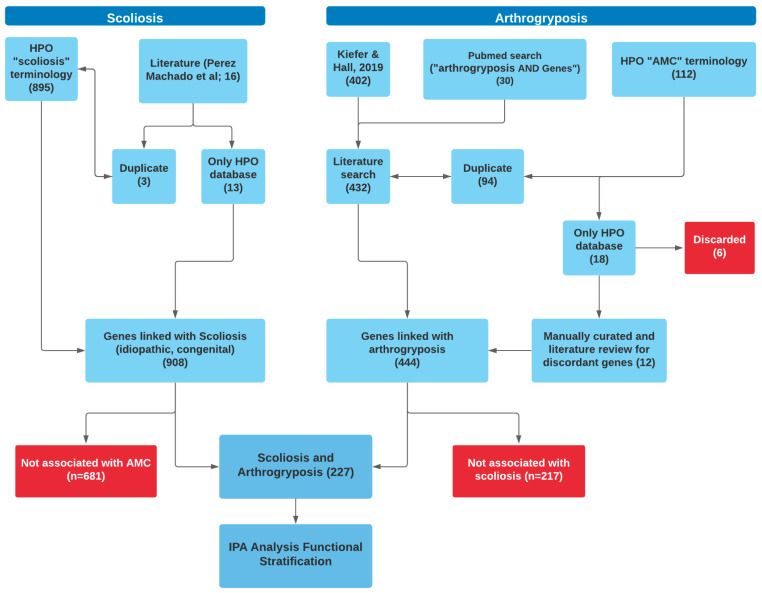
Flow chart diagram used in the current study to identify relevant genes associated with both scoliosis and arthrogryposis multiplex congenita (AMC).

**Figure 2 genes-12-01052-f002:**
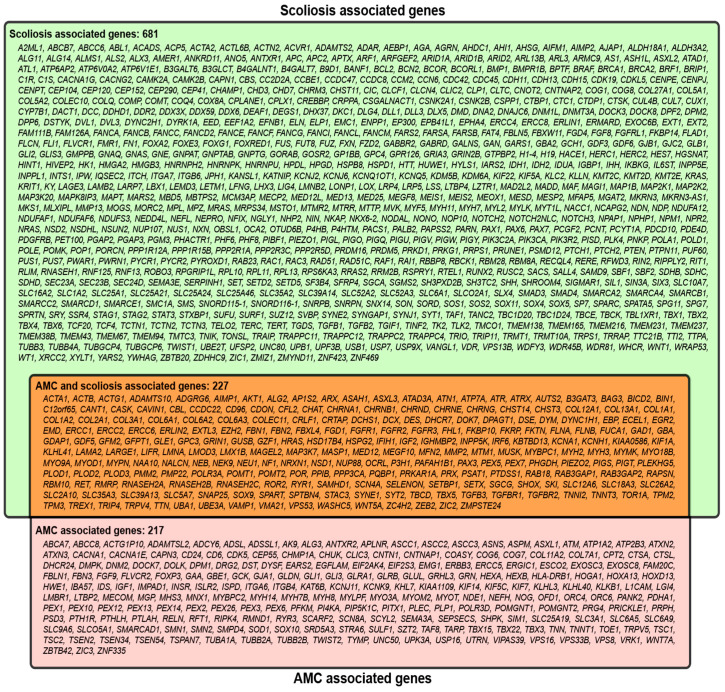
Scoliosis-associated genes are indicated in the green box, AMC genes are indicated in the red box, and scoliosis and AMC genes are in the orange region.

**Figure 3 genes-12-01052-f003:**
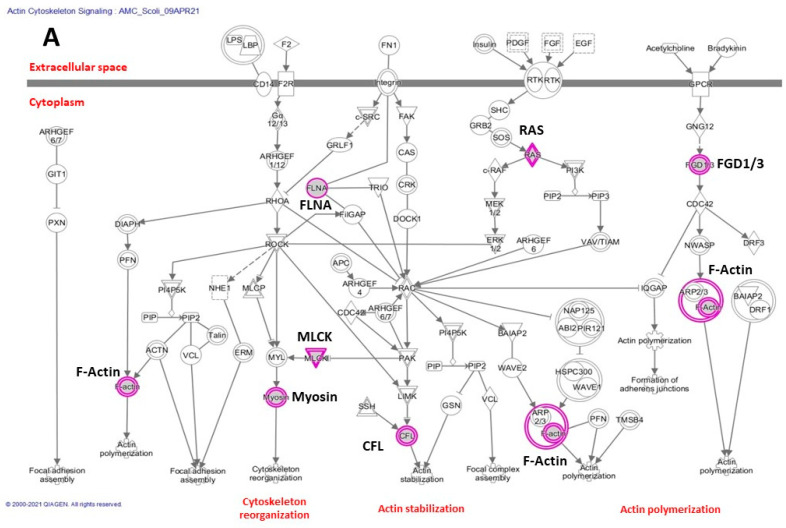
Two examples representative of pathway analysis of the final gene set by Ingenuity Pathway Analysis (QIAGEN). (**A**) Actin cytoskeleton signaling canonical pathway. Genes that are part of the AMC-SC list are shown in purple. (**B**) Highest score pathway predicted by IPA connecting the largest number of gene list members. Solid lines indicate direct interaction between genes while dotted lines symbolize indirect connection. Molecule shapes indicate gene functions, legends can be found here: https://qiagen.secure.force.com/KnowledgeBase/KnowledgeIPAPage?id=kA41i000000L5rTCAS (accessed on Wednesday 13 January 2021).

**Figure 4 genes-12-01052-f004:**
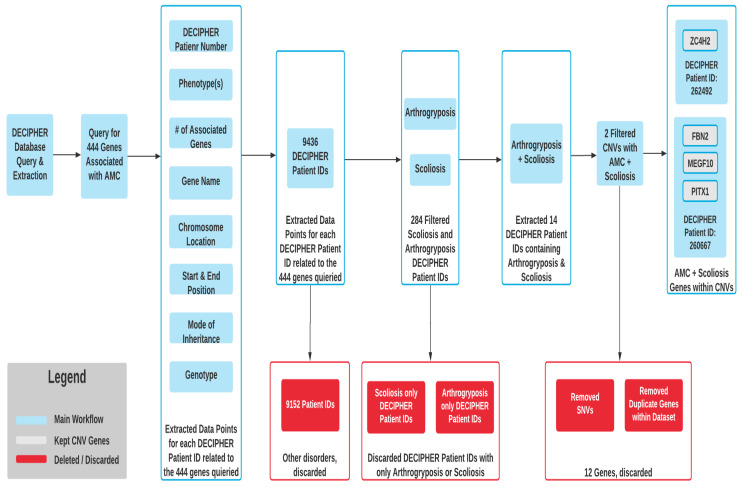
DECIPHER Gene Extraction Workflow.

**Table 1 genes-12-01052-t001:** The 227 Genes Associated with AMC-SC Stratified by Function.

**Homeostasis:** mechanisms include genes associated with early-onset nuclear DNA excision/repair disorders (*ERCC1*, *ERCC2,* and *ERCC6*) [13]. Monoallelic mutations in ERCC1 are associated with cerebro-oculo-facio-skeletal syndrome. *ERLIN1* encodes for a lipid raft-associated protein localized to the mitochondrion and nuclear envelope, and is a component of the *ERLIN1/ERLIN2* complex. The complex mediates the endoplasmic reticulum-associated degradation of inositol 1,4,5-trisphosphate receptors (ITPRs) which are important in calcium homeostasis [14]. *BAG3, BIN1, ERCC1, ERCC2, ERCC6, ERLIN2, SELENON*
**Cytoskeleton:** matrix proteins involved in the sarcomere such as nebulin, a giant protein of thick and thin filaments of striated muscle, encoded by *NEB*. Mutations in NEB are responsible for the majority of cases of nemaline myopathy [15] which can be diagnosed by Gomori trichrome staining on a muscle biopsy or by electron microscopic preparation. *ACTA1*, a member of the cytoskeletal grouping, encodes the principal skeletal muscle isoform of adult skeletal muscle, alpha-actin. Residing in the core of the thin filament of the sarcomere, it assists in the generation of muscle contraction [16] *ACTA1, ACTB, ACTG1, AP1S2, COL12A1, COL13A1, COL1A1, COL1A2, COL2A1, COL3A1, COL6A1, COL6A2, COL6A3, COLEC11, DCX, DES, DYNC1H1, EMD, FBN1, FBN2, FLNA, FLNB, HSPG2, LMNA, NEB, SPTBN4, SYNE1, TBCD*
**Extra Cellular Matrix:** Extracellular matrix (ECM) protein-associated genes include *ADAMTS10* and *DCHS1. ADAMTS10* is a zinc-dependent protease composed of one cysteine-rich domain, and five thrombospondin type 1 (THBS1) repeats and plays an important role in the formation of the extracellular matrix [17]. *DHCS1* is a member of the protocadherin superfamily and encodes a transmembrane cell adhesion molecule responsible for apical anchoring in the brain [18]. *ADAMTS10, CDON, DCHS1, MMP2, RAPSN*
**Signal Transduction:** Promotes signaling within a cell via enzyme network cascades to generate precise and appropriate physiologic responses, particularly in skeletal development. *FGFR3* codes for an important tyrosine kinase signal transducer in chondrocytes, functioning to attenuate cartilage growth. FGFR 1–4 transmit at least 18 different fibroblast growth factor (FGF) ligands, therefore, exhibiting a variety of physiological functions [19]. *GDF5* fulfills important functions with respect to bone and muscle [20]. Through its high affinity for BMPR1B, GDF5 positively regulates chondrogenesis, leading to SMAD signal transduction [21]. Through NOG mediated interaction, GDF5 paradoxically also negatively regulates chondrogenesis. *ADGRG6, CAVIN1, CCDC22, CD96, CFL2, CRLF1, CRTAP, DOK7, EBP, FGFR1, FGFR2, FGFR3, GDF5, IFIH1, KIAA0586, MAGEL2, NF1, PEX5, PEX7, PMP22, RAB3GAP1, RAB3GAP2, STAC3, WNT5A, KBTBD13*
**Proto-oncogenes:** Proto-oncogenes act to facilitate dysregulated cell growth and differentiation. Mutations in *HRAS* are associated with Costello syndrome, characterized by distinct facial features, papilloma of the face, cardiac anomalies, growth restriction, developmental delays, and tumor predisposition. An *HRAS* mutation was identified in an infant with features of Costello syndrome and distal arthrogryposis [22]. *AKT1, CBL, HRAS, RAB18, RET, SKI*
**Enzyme:** Account for the largest category of genes identified through IPA analysis. 7-dehydrocholesterol reductase (DHCR7) encodes the penultimate step in the cholesterol biosynthetic pathway. Smith-Lemli-Opitz Syndrome is an autosomal recessive disorder caused by an inherited deficiency of DHCR7 which is associated with a variety of birth defects, joint contractures, and intellectual disability [23]. *UBE3* which encodes E3 ubiquitin-protein ligase, a maternally expressed imprinted E3 ubiquitin-protein ligase expressed mainly in the brain, is an integral part of the ubiquitin protein degradation system. Angelman syndrome, characterized by severe cognitive impairment, seizures, an ataxic puppet-like gait, and paroxysms of laughter, is caused by an absence of expression of maternal *UBE3A* [24]. *ALG2, ASAH1, B3GAT3, CANT1, CHAT, CHST14, CHST3, DHCR7, DPAGT1, DSE, ECEL1, EXTL3, EZH2, FBXL4, FKRP, FUCA1, GAD1, GBA, GFPT1, GUSB, HSD17B4, INPP5K, LARGE1, MASP1, MTM1, NAA10, NEU1, OCRL, P3H1, PAFAH1B1, PHGDH, PLOD1, PLOD2, PLOD3, PMM2, POLR3A, POMT1, POMT2, POR, PPIB, PPP3CA, PSAT1, PTDSS1, RNASEH2A, RNASEH2B, RNASEH2C, SAMHD1, TOR1A, TREX1, UBA1, UBE3A, ZMPSTE24*
**Transcription Factor/regulation:** Transcription factors have a pivotal role in the regulation of genes associated with limb and muscle development. T-Box Transcription Factor 5 (*TBX5*) mutations are associated with Holt Oram syndrome characterized by upper limb defects and cardiac malformations [25,26]. *TRIP4* encodes ASC-1, a transcription co-activator. Infants with *TRIP4* mutations present with a congenital muscular dystrophy and respiratory failure. Muscle biopsy shows decreased mitochondria and sarcomere disorganization [27]. *ARX, ASXL3, ATN1, ATRX, AUTS2, EGR2, FGD1, GZF1, IGHMBP2, IRF6, LMX1B, MED12, NSD1, PAX3, PLEKHG5, PQBP1, RBM10, SETBP1, SETX, SHOX, SOX9, TBX5, TRIP4, ZC4H2, ZEB2, ZIC2*
**Mitochondria:** Mitochondria are depended upon highly by the brain and skeletal muscle tissues for energy. Ganglioside Differentiation Associated Protein 1 (*GDAP1*) encodes a mitochondrial protein postulated to play a role in signal transduction in the brain. Mutations in *GDAP1* are associated with various subtypes of the hereditary and sensory-motor neuropathy disease Charcot Marie Tooth (CMT), including an autosomal recessive intermediate type [28,29,30,31,32]. *RMRP* codes for non-coding RNA involved in mitochondrial DNA replication through the encoding of a mitochondrial RNA processing endonuclease which cleaves mitochondrial RNA at a priming site necessary for mitochondrial DNA replication. Mutations in *RMRP* are associated with cartilage-hair hypoplasia [33]. *RMRP* is essential for early murine development [34]. *ATAD3A, C12orf65, GDAP1, GFM2, MFN2, RMRP, SPAR*
**Membrane Receptor/Ion Channel:** Membrane receptor and ion channels is the second largest group of affected genes leading to AMC-SC. *CHRNA1* (cholinergic receptor nicotinic receptor alpha 1 subunit 1) is one of 5 subunits of the acetylcholine receptor (*AChR*). This gene encodes an alpha subunit and functions as part of acetylcholine binding and channel. Mutations in *CHRNA1* are associated with lethal multiple pterygium syndrome, characterized by the presence of multiple pterygia, intrauterine growth retardation, and flexion contractures resulting in severe arthrogryposis and fetal akinesia [35]. *PIEZO2* is postulated to function as an integral part of mechanically activated cation channel in somatosensory neurons through establishing connections between mechanical forces and biological signals. Mutations in *PIEZO2* are associated with distal arthrogryposis type 5, Gordon syndrome, and Marden–Walker syndrome [36]. *ATP7A, CHRNA1, CHRNB1, CHRND, CHRNE, CHRNG, GPC3, GRIN1, KCNA1, KCNH1, MEGF10, NALCN, NRXN1, NUP88, PIEZO2, PIGS, PIGT, ROR2, RYR1, SCN4A, SGCG, SLC12A6, SLC18A3, SLC26A2, SLC2A10, SLC35A3, SLC39A13, SLC5A7, SNAP25, SYT2, TGFBR1, TGFBR2, TRPV4, VAMP1, WASHC5*
**Kinase:** Kinases phosphorylate target molecules for activation or inactivation. ATR encodes a serine/threonine kinase and halts cell cycling entry upon DNA stress to enable DNA repair [37]. Compound heterozygous mutations in ATR are associated with Seckel syndrome characterized by dwarfism, microcephaly, and cognitive impairment [38]. MAP3K7 mediates cellular transduction in response to environmental changes through association with interleukin receptor (ILR1). Through the cytokine IL-1 mediated interaction with the hypothalamic IL-1 receptor, the hypothalamo-pituitary-adrenocortical axis and sympathetic nervous system pathways suppressing bone formation are activated [39]. Fronto-metaphyseal dysplasia, a progressive sclerosing skeletal dysplasia characterized by small bone undermodeling, supraorbital hyperostosis, large and small joint contractures as well as developmental abnormalities, of the cardiorespiratory system and the genitourinary tract is associated with *MAP3K7* mutations [40]. *ATR, CASK, MAP3K7, MUSK, NEK9, PRKAR1A*
**Intracellular transport:** Intracellular transport proteins are structural proteins that facilitate the movement of vesicles and substances within a cell. *BICD2* codes for a structural protein functioning as an intracellular adaptor for the dynein motor complex, linking it to various cargos. Through the stabilization of the interaction between dynein and dynactin, the movement of dynein is facilitated along the microtubule [41]. Mono-allelic mutations in *BICD2* cause congenital spinal muscular atrophy [42]. *GLE1* is postulated to act as a terminal step in the transport of mature messenger RNA messages from the nucleus to the cytoplasm. Bi-allelic mutations in *GLE1* are associated with a lethal congenital contracture syndrome characterized by fetal hydrops, degeneration of anterior horn cells, and congenital contractures [43]. *BICD2, DYM, FKBP10, GLE1, KIF1A, VPS53*
**Structural:** Structural proteins provide the framework for a cell or complex of cells. The *LAMA2* gene encodes laminin-2 or merosin, a major component of the extrasynaptic membrane of muscle cell basement membrane. Laminin-211 binds to the glycosylated residues of alpha-dystroglycan (*DAG1*) in skeletal muscle fibers [44]. Bi-allelic mutations in *LAMA2* are associated merosin-deficient congenital muscular dystrophy. Affected patients have hypotonia, joint contractures and may develop scoliosis. Myosin, the major contractile protein in muscle, is composed of two heavy chains and two light chains. *MYH3* encodes the embryonic myosin heavy chain 3. *MYH3* mutations appear to reside near a groove that is part of the myosin head and are associated with distal arthrogryposis type 1 in which contractures are limited to distal joints, Freeman –Sheldon, Sheldon -Hall syndromes [45]. Affected patients with Freeman Sheldon and Sheldon Hall syndromes have distal joint contractures, characteristic facial features and may develop scoliosis. *MYH3* mutations are postulated to cause structural changes in myosin that potentially alter myosin domain-domain interactions during ATP catalysis or affect nucleotide-binding site conformation. *FHL1, FKTN, KLHL41, LAMA2, LMOD3, MYBPC1, MYH2, MYH3, MYMK, MYO18B, MYO9A, MYOD1, MYPN, PRX, TNNI2, TNNT3, TPM2, TPM3, TTN, VMA21*

**Table 2 genes-12-01052-t002:** CNV Associated with Scoliosis and Arthrogryposis.

Title	Title	Altered Genes		
	Phenotype	Gene 1	Gene 2	Gene 3
Patient 1	Cleft palate, crumpled ear, distal arthrogryposis, intellectual disability, micrognathia, scoliosis, syringomyelia; mild pulmonary stenosis	*FBN2* (Chr5, de novo, loss, heterozygous) *	*MEGF10* (Ch5, de novo, loss, heterozygous) *	*PITX1* (Ch5, de novo, loss, heterozygous) *
Patient 2	AMC, dysphagia, dystonia, global developmental delay, laryngomalacia, thoracolumbar scoliosis	*ZC4H2* (ChX, de novo, loss, het) *	N/A	N/A

Patient 1 (DECIPHER #260667): https://decipher.sanger.ac.uk/patient/260667/overview/general (accessed on Monday 28 June 2021). Deletion chr 5: Start position 125286403, length: 10815843, contains 133 genes. Patient 2 (DECIPHER #262492): https://decipher.sanger.ac.uk/patient/262492/overview/general (accessed on Monday 28 June 2021). Deletion chr X: Start position: 64954439, length: 233145, contains 1 gene. * For each gene the following information is provided between parentheses: chromosome number (chr), CNV inheritance (de novo, heterozygous) and category (gain/loss).

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
