# Peer review of "A Genomic Approach to Delineating the Occurrence of Scoliosis in Arthrogryposis Multiplex Congenita"

_genes, 2021, doi:10.3390/genes12071052_

Round 1

Reviewer 1 Report

The idea of the authors to use in silica tools to reveal common genes and pathways in the molecular pathophysiology of arthrogryposis and scoliosis is novel. Their work is presented with clarity.

My suggestion is additional analysis of comparison between pathways associated with arthrogryposis without scoliosis, scoliosis without arthrogryposis and pathways that are associated with both arthrogryposis and scoliosis.

Are the pathways presented in the discussion involved in other unrelated  congenital anomalies? And if yes, are there distinguishing characteristics of the pathways associated with arthrogryposis and scoliosis? May be addressing these questions will enrich the article and the conclusions.

Author Response

Reviewer 1

The authors delineate pathways involved in the etiology of arthrogryposis multiplex and congenital scoliosis by using a combined approach of literature review and human phenotype ontology to identify genes involved in both phenotypes and to then analyse their pathways using IPA. This adds to the current literature since in some AMC phenotypes scoliosis is present, in others not and there is little understanding of the mechanism. Although gene ontology enrichment analysis would be a more powerful tool, the genomic approach presented is valid to identify candidate pathways. The paper is well written. I nevertheless have a few questions and comments.

  1. The authors should comment if the identification of genes associated with AMC (2.1) also included genes which are already known to cause a scoliosis-AMC phenotype. Including those in the analysis, depending on how many, may lead to ascertainment bias in the final combined gene pool for overlapping genes. Known scoliosis-AMC genes should be regarded separately or at least their contribution to the overlap group should be accounted for.

Based on our assumption of what the reviewer is asking, and particularly pertaining to the methods used for gene identification, our impression is that we could not have been able to reveal new AMC – scoliosis associations for a given, already known, gene. Indeed each gene we searched for was either associated with scoliosis, AMC or both before the analysis. We do not believe there was any ascertainment bias in the experimental design of this study.

The overreaching aim of this manuscript was to give an overview of the overlapping genes that are found in AMC and associated with scoliosis and vice versa in order to identify common intracellular pathways and functions underlying these conditions.

Since the key entries, namely AMC and scoliosis, are clinical conditions, an in depth literature review on every publication on AMC and scoliosis would be necessary to separate known genes that associate AMC + scoliosis from not yet known. This in depth information is not necessarily captured by the bioinformatics approach.

We have clarified this point in the Discussion.

  1. The authors identify a number of pathways common to AMC and scoliosis. It would be also of great interest to know if and which pathways differ between both groups.

This suggestion is similar to reviewer 2’s second suggestion. We think this could be an interesting work for a second manuscript given the timeline for revision.

  1.  Last paragraph of results section on CNVs: The authors should mention if they think that the decipher patients have contiguous gene deletion syndromes and proved a statement about haploinsufficieny scores and (loss of) function mechanisms of the genes identified in the CNVs.

The first patient has a 10.82Mb heterozygous deletion containing 133 genes resulting in a contiguous gene deletion syndrome. This deletion has been documented with a haploinsufficiency score of 50.51, i.e. a high likelihood of causing a loss of function. We suspect this patient has a contiguous gene syndrome. Additional evidence of similar cases with this deletion would be necessary to support this hypothesis. From a molecular perspective, FBN2 intragenic deletions or splice site mutations have been published on some occasions associated with contractural congenital arachnodactyly (Meerschaut et al, 2019). Rare nonsense mutations are present in the ClinVar database and reported as pathogenic. It is nevertheless not mentioned in detail whether or not these latter mutations are associated with an AMC phenotype. Therefore we can neither exclude nor confirm a link between FBN2 haploinsufficiency and AMC. One microdeletion involving only PITX1 has been associated in one family with clubfoot over three generations (Alvarado et al, 2011). Other nonsense variants (2) have only been mentioned in ClinVar. As a consequence, it is not possible with our current data, to determine more precisely if and if so, in which proportion each gene contributes the AMC and/ or scoliosis phenotype.

The second patient reported has a microdeletion that encompasses only the ZC4H2 gene. Loss of function mutations have been described to be pathogenic on several occasions in ZC4H2 (Hirata et al, 2013; May et al, 2015; Okubo et al, 2016; Zanzotera et al, 2017; Frints et al, 2019), underlining the pathogenic nature of ZC4H2 haploinsufficiency.

  1. The discussion reads much too long. It is a summary description and enumerations of the overlapping pathways identified and the conditions associated. This should be significantly shortened to a more abstract discussion on key findings and conclusions. Details on each pathway could e.g. be given in a table, which also would be much easier to follow.

Our idea was to give especially the non-geneticist reader an example of at least some of the pathways identified through the IPA analysis and their interactions. This represents finally only a part of the discussion. To make the discussion shorter we have removed the descriptions of pathways and genes and added this to Table 2.

  1. Figure 1,2 and 4 should be improved in terms of readability.

We have improved Figures 1, 2 and 4 for readability.

Reviewer 2 Report

The authors delineate pathways involved in the etiology of arthrogryposis multiplex and congenital scoliosis by using a combined approach of literature review and human phenotype ontology to identify genes involved in both phenotypes and to then analyse their pathways using IPA. This adds to the current literature since in some AMC phenotypes scoliosis is present, in others not and there is little understanding of the mechanism. Although gene ontology enrichment analysis would be a more powerful tool, the genomic approach presented is valid to identify candidate pathways. The paper is well written. I nevertheless have a few questions and comments.

  1. The authors should comment if the identification of genes associated with AMC (2.1) also included genes which are already known to cause a scoliosis-AMC phenotype. Including those in the analysis, depending on how many, may lead to ascertainment bias in the final combined gene pool for overlapping genes. Known scoliosis-AMC genes should be regarded separately or at least their contribution to the overlap group should be accounted for.
  2. The authors identify a number of pathways common to AMC and scoliosis. It would be also of great interest to know if and which pathways differ between both groups. 
  3.  Last paragraph of results section on CNVs: The authors should mention if they think that the decipher patients have contiguous gene geletion syndromes and proved a statement about haploinsufficieny scores and (loss of)function mechanisms of the genes identified in the CNVs.
  4. The discussion reads much too long. It is a summary description and enumerations of the overlapping pathways identified and the conditions associated. This should be significantly shortened to a more abstract discussion on key findings and conclusions. Details on each pathway could e.g. be given in a table, which also would be much easier to follow.
  5. Figure 1,2 and 4 should be improved in terms of readability. 

Author Response

Dear Reviewers:

We thank you for your thoughtful comments regarding our manuscript.  Please see our responses below.

Sincerely,

Klaus Dieterich M.D. Ph.D.

Philip F. Giampietro M.D. Ph.D.

Reviewer 2:

The idea of the authors to use in silica tools to reveal common genes and pathways in the molecular pathophysiology of arthrogryposis and scoliosis is novel. Their work is presented with clarity.

We thank the reviewer for this kind commentary.

My suggestion is additional analysis of comparison between pathways associated with arthrogryposis without scoliosis, scoliosis without arthrogryposis and pathways that are associated with both arthrogryposis and scoliosis.

The focus of this paper was to identify genes/pathways that have an AMC-scoliosis phenotype.  While an interesting comparison, this would require a lot of work and deviate from the primary goal of this paper, “to characterize the genetics of the AMC types that have a strong association with scoliosis”.

Nevertheless we thank the reviewer for this suggestion. Given the timeline, any further analysis would unfortunately take much longer than the time for revision of this manuscript. We believe this additional analysis would be very interesting though and we keep this idea in mind for another manuscript.

Are the pathways presented in the discussion involved in other unrelated congenital anomalies? And if yes, are there distinguishing characteristics of the pathways associated with arthrogryposis and scoliosis? May be addressing these questions will enrich the article and the conclusions.

We have added the following statement to the manuscript:

IPA analysis did identify other potential disorders and conditions that may be attributed to alterations in genes which are members of pathways in which the 227 genes identified with AMC and scoliosis. These include skeletal, muscular, limb defects and cognitive disability. Further validation would require a more in-depth analysis, which is not the focus of this paper.